# Responses of Soybean Genotypes to Different Nitrogen and Phosphorus Sources: Impacts on Yield Components, Seed Yield, and Seed Protein

**DOI:** 10.3390/plants11030298

**Published:** 2022-01-24

**Authors:** Raby Nget, Edna A. Aguilar, Pompe C. Sta. Cruz, Consorcia E. Reaño, Pearl B. Sanchez, Manuel R. Reyes, P. V. Vara Prasad

**Affiliations:** 1Faculty of Agronomy, Royal University of Agriculture, Dangkor District, Phnom Penh P.O. Box 2696, Cambodia; 2Institute of Crop Science, College of Agriculture and Food Science, University of the Philippines, Los Baños 4032, Philippines; eaaguilar1@up.edu.ph (E.A.A.); pcstacruz@up.edu.ph (P.C.S.C.); cereano@up.edu.ph (C.E.R.); 3Agricultural Systems Institute, College of Agriculture and Food Science, University of the Philippines, Los Baños 4031, Philippines; pbsanchez1@up.edu.ph; 4Sustainable Intensification Innovation Lab, Department of Agronomy, Kansas State University, Manhattan, KS 66506, USA; mannyreyes@ksu.edu

**Keywords:** soybean, nitrogen, phosphorous, yield, yield components, seed protein

## Abstract

Soybean (*Glycine max* [L.] Merr) is an important crop, as both food for humans and feed for livestock in Cambodia, but the yields are low, due to use of low yielding genotypes and limited use of inputs. This study aimed to investigate the effects of different genotypes and different N and P fertilizer sources on growth, seed yield, and seed protein of soybean. Five genotypes (Sbung, Hongry, 98C81, ACS1, and Kaiabi) were grown under different N (90 kg N ha^−1^ from urea, nano-n, Nitroplus, and without N application as control) and P fertilizers (50 kg P ha^−1^ from Inorganic P, nano-p, Mykovam, and without P application as control) in two screenhouse experiments. Shoot, root, and nodulation traits, as well as seed yield and protein, were significantly affected by genotypes and different N and/or P fertilizer sources. Notably, while genotypes Sbung and 98C81 showed the highest yields among the genotypes, regardless of different N and P fertilizers, only Sbung had the highest seed protein. The application of different N and/or P sources significantly increased seed yield, compared to non-N and -P application. Specifically, seed yield and seed protein were higher when treated with nano-n or urea, and number of nodules, root biomass, and nodule dry weight was increased with Nitroplus, whereas seed yield and protein were higher with nano-p, Mykovam, and inorganic P. Application rate of 90 kg N ha^−1^ and 50 kg P ha^−1^ produced higher seed yield and its components and seed protein. We conclude that N and P application as well as Nitroplus inoculation can help increase seed yield of soybean in Cambodia.

## 1. Introduction

Soybean (*Glycine max* L. Merrill) is one of the most important legume crops and essential source for high quality protein, edible oil, and micronutrients for human food and livestock feed around the world [1]. Soybean protein is rich in valuable amino acid lysine (5%), which is deficient in most of the cereals [2]. Soybean, like other legumes, fixes atmospheric nitrogen (N_2_), through association with gram-negative bacteria species of the genera *Bradyrhizobium* and *Sinorhizobium* [3]. Since there is no single organic or inorganic source to meet the needs for all plant nutrients, integrated use of all nutrient sources namely, organic (e.g., compost, crop residues, and manures), inorganic chemical fertilizers, and biofertilizer needs careful attention and consideration. The leguminous crop requirement of N is substantially fulfilled from symbiotic N_2_ fixation through *Rhizobium*, i.e., 120–150 kg N ha^−1^ [4]. In addition, legumes leave about 30–40 kg N ha^−1^ for the succeeding cereal crops [5]. The macronutrient N encourages aboveground vegetative growth and partitioning. It is an essential component of amino acids and related proteins, as well as many compounds, including chlorophyll and enzymes, vital for photosynthesis and plant growth processes. It is also required for carbohydrate metabolisms within plants, root growth stimulation, and uptake of other nutrients. The phosphorus (P), which is also a macro element in agricultural production systems, is in greatest demand in soybean plants during pod and seed development, as more than 60% of P ends up in the pod and seeds. It plays an important role in photosynthesis, transfer of energy, nutrient movement, carbohydrate metabolism, and biological nitrogen fixation.

The global average yield of soybean is about 2.8 t ha^−1^, whereas it is only 1.6 t ha^−1^ in Cambodia [6,7]. This low soybean yield in Cambodia is due to the use of traditional varieties, poor crop establishment, inappropriate planting depth and plant population, low soil fertility, and lack of effective N fixation [8]. Moreover, a large proportion of arable soil in Cambodia has undergone soil degradation, leading to nutrient depletion [9]; thus, N and P are deficient, which often limit crop production [10]. Limited availability of P could adversely impact biological N fixation [11,12], and the deficiency of these macronutrients is the major constraint in soybean production, since it affects growth, yield, nodulation, and N fixation [13], as well as the impacts of seed composition (protein, oil, fatty acids, and sugars) and seed nutrients (B, P, and Fe) of the plant [14].

The response of the crop plants, including soybean, to nutrients depends on the source of organic or inorganic fertilizers. Most farmers use traditional sources of N and/or P, such as urea, single super phosphate, diammonium phosphate, or mixed fertilizers. When these fertilizers are not applied appropriately it can be lost, due to environmental conditions and poor management practices. Usage of inorganic fertilizers in Cambodia is low, due to availability, cost, or high losses; farmers are encouraged to use alternative sources to enhance soil fertility and increase crop yields. In recent years, there have been new sources of nutrients, which can minimize nutrient losses, can increase absorption by plants, and are more bioavailable to the plants. One such source is nano-fertilizers, which have a large surface area and slow and steady release of nutrients, both of which make them highly suitable for use in modern agriculture [15,16]. According to some recent estimation, soybean responds well to about 30% of attained crop productivity with the use of different sources of fertilizers, namely nano-fertilizer, biofertilizer, inorganic fertilizer, and *rhizobium* [17]. Okon and Labandera-Gonsales [18] found that plant-growth promoting rhizobacteria (PGPR), such as Nitroplus, has a success for inoculation of 60–70%, with significant yield increase in sorghum (*Sorghum bicolor* L. Moench) or millet (*Pennisetum glaucum* L. R. Br.). Application of PGPRsstimulated density and length of root hairs, rate of appearance of lateral roots, and root surface area, resulting to increase N and P uptake, reduction in losses, and promotion of nodulation that can enhance biological N fixation [19].

In Cambodia, like many developing countries, agriculture lands are predominately occupied by smallholder farmers, who have limited resources and capability (knowledge and technology) to manage the soils efficiently and optimize crop production [9]. Salvagiotti et al. [20] reported that the use of the high-yielding genotypes, in addition to adoption of appropriate *Rhizobium* inoculant application with N and P, has the potential to improve soil fertility and soybean yield. This knowledge on soybean, in combination with these factors, limits productivity, especially by resource-poor smallholder farmers. Although cultivation of soybean with different sources of N and P is seen as a good option to increase availability of the nutrients in the soil and enhance production, the effects of N and P fertilization on growth, yield, and yield component and nodulation of soybean are not well investigated.

Furthermore, the responses to different N and P fertilizer sources may vary with soybean genotypes with differential yield and seed quality. Despite large genetic variability in soybean limited studies have been conducted on the responses of various genotypes to different sources of N and P fertilizers, as well as their influence on growth, yield, and protein. Such understanding is needed to improve soybean yield in Cambodia. The objective of the study was to investigate the effects of different N and P fertilizer sources on the growth, yield, and seed protein of five soybean genotypes in Cambodia.

## 2. Results

### 2.1. Experiment 1: Effects of Soybean Genotypes to N Fertilizer Sources on Seed Yield and Its Components and Seed Protein

#### 2.1.1. Shoot Growth Traits

Table 1 presents the ANOVA on the effects of genotypes (G), N sources (N), and their interactions (GxN) on plant height, days to flowering (DF), number of branches, and stem biomass. The analysis showed consistently significant effects of genotype on all the traits, as well as effects of N sources and GxN, except for the number of branches. Significant differences among the genotypes (*p* < 0.05) for plant height at 28, 42, and 56 days and DF were observed. Genotype ACS1 consistently showed the highest plant height at different days (49.37, 91.94, and 94.43 cm, respectively) but had the longest DF (41.03), compared to the other genotypes that had the earliest DF (35.18–36.09). Moreover, genotype Sbung had the highest stem biomass (5.39 g plant^−1^), whereas genotype ACS1 showed the lowest stem biomass. Among the N sources, 90 kg N ha^−1^ urea application yielded the highest plant height at 28 days after growing (46.08 cm), regardless of genotypes. Urea fertilizer also showed the highest effect in stem biomass (5.50 g plant^−1^), followed by nano-n (4.92 g plant^−1^) and Nitroplus (4.86 g plant^−1^), whereas non-application of N had the lowest effect (4.54 g plant^−1^). Generally, the N source that showed the highest positive effects to DF and stem biomass was urea, although the effects of the other N sources (Nitroplus, nano-n) on shoot growth were greater than non-N application.

#### 2.1.2. Chlorophyll Index

The chlorophyll index of the leaves, estimated by the SPAD values, are presented in Table 2. The higher the SPAD value, the higher the chlorophyll index. There were significant effects of the genotype and N sources (*p* < 0.01) but not interaction (*p* > 0.05). The genotype 98C81 had the highest leaf chlorophyll index, consistently observed from 28 to 70 days after sowing, ranging from 35.6 to 43.5. Application of 90 kg N ha^−1^ of nano-n, urea, and Nitroplus yielded the higher leaf chlorophyll index, obtained at 28 and 56 days after sowing, compared to the control. At 70 days, the leaf chlorophyll index was similar among nano-n, urea, and Nitroplus (43.5, 42.8, and 42.7, respectively) but significantly higher than the control. This suggests that application of N of different sources increased the chlorophyll index of soybean, compared to non-N application.

#### 2.1.3. Root and Nodulation Traits

The result of the effects of genotypes and different N sources on root biomass, number of nodules, nodule dry weight, and shoot-to-root ratio are presented in Table 3. The analysis showed consistently significant effects of genotypes and N fertilizer sources on all the below-ground traits, as well as GxN interactions, except for the root biomass. When averaged across N sources, genotypes Sbung, 98C81, and Kaiabi, produced significantly higher root biomass (7.51, 7.25, and 7.04 g plant^−1^, respectively), compared to genotypes Hongry and ACS1 (5.98 and 5.59 g plant^−1^, respectively). In the case of number of nodules and nodule dry weight per plant, genotype Sbung consistently showed the highest in both traits (34.25 nodules and 0.42 g, respectively), whereas genotypes Sbung, Hongry, and ACS1 showed the highest shoot-to-root ratio (2.54, 2.51, and 2.43, respectively). Among the N fertilizer sources, inoculation of Nitroplus consistently showed the greatest effect on all the root traits examined, which included highest root biomass (8.78 g plant^−1^), number of nodules per plant (32.58), nodule dry weight was obtained at urea and Nitroplus (0.36 and 0.35 g plant^−1^), and shoot-to-root ratio was obtained at urea, nano-n, and Nitroplus (2.48, 2.45 and 2.34, respectively). The other N sources, urea and nano-n, showed the effects lower than Nitroplus, but their effects were higher when compared to the control. Generally, application of N from different sources showed positive effects on root traits, as compared to without N application.

#### 2.1.4. Yield Traits, Harvest Index, and Seed Protein

Table 4 shows the effects of genotypes and different N sources on the number of filled pods, total pods per plant, and number of seeds per pod 100-grain weight, seed yield per plant, harvest index, and seed protein. There were significant effects of genotypes (*p* < 0.01) and N fertilizer sources (*p* < 0.01) on all the traits. When means were obtained across N sources, genotypes Sbung and 98C81 had the highest filled pods and total pods, 100-grain weight, seed weight, and harvest index, whereas the genotypes Kaiabi, ACS1, and Hongry had the lowest. All genotypes, except Hongry, produced similar number of seeds per pod. However, for seed protein, only genotype Sbung produced the highest (36.9%), followed by genotypes 98C81 (35.9%) and Kaiabi (35.7%). When averaged, regardless of genotypes, to obtain the effects of fertilizers, 90 kg N ha^−1^ urea application showed the highest number of filled and total pods (31.9 and 34.19, respectively), while the control had the lowest (27.46 and 30.56). Application of all different N sources, likewise, had significant effects on the number of seeds per and 100-grain weight, when compared to control. For seed yield, harvest index, and seed protein, only urea and nano-n had effects, compared to Nitroplus and control. There were significant differences in seed weight and harvest index between urea and Nitroplus. Among the traits, interaction (GxN) effects were significant (*p* < 0.05) only in seed protein. Overall, both genotypes Sbung and 98C81 showed the highest seed yield, but only Sbung had the highest seed protein. On the other hand, urea, nano-n, and inoculation of Nitroplus increased yield, as compared to non-application of N.

#### 2.1.5. Correlation Analyses among Yield and Yield Components

Highly significant and positive correlations between seed yield and 100-grain weight (r = 0.53), seed protein (r = 0.50), and total number of pods per plant (r = 0.39), as well as 100-seed weight and total number of pods per plant (r = 0.50) were found (Table 5). There was a significant correlation between DF and plant height (r = 0.48), seed protein, and 100-grain weight (r = 0.49), and seed protein and number of pods per plant (r = 0.59). The results indicate that seed yield, in all treatments, was positively and significantly correlated with 100-grain weight and total number of pods per plant, which showed the attribution of different seed yield. Nodule characteristics were also correlated significantly and positively with each yield attributes, except shoot biomass per plant. The yield of soybean is a dependent variable, wherein it depends upon all yield traits, seed protein, root, and nodulation.

#### 2.1.6. Path Analysis

Path analysis for direct and indirect effects of soybean on seed yield is presented in Table 6. It is indicated positive associations between plant heights, number of branches, root biomass, shoot biomass number of pod per plant, 100-grain weight, and seed yield. The results of the path analysis confirmed the correlation analysis results. However, among these traits, 100-grain weight (0.306) has the most positive direct effect on soybean yield, followed by number of pods per plant (0.182) (Table 6). This information can be utilized in determining selection criteria in soybean growing under N sources.

### 2.2. Experiment 2: Effects of Soybean Genotypes to P Fertilizer Sources on Seed Yield and Its Components and Seed Protein

#### 2.2.1. Shoot Growth Traits

The ANOVA results of the effects of genotypes and different P sources on plant height, DF, stem biomass, and number of branches is presented in Table 7. There were significant effects of genotype (*p* < 0.01) and P fertilizer sources (*p* < 0.05) for plant height at 28, 42, and 56 days, as well as DF and stem biomass, but no effects in the number of branches. The genotype ACS1 consistently showed the greatest height at different days (46.07, 88.12, and 90.61 cm, respectively) but had the longest DF (41.55), compared to the other genotypes, Hongry, that had significant earliest DF (34.32). Whereas genotypes Sbung and Hongry showed the highest stem biomass, regardless of P treatments. Among P sources, 50 kg P ha^−1^ of inorganic P yielded the highest plant height at different days (80.91 cm at 56 days) and earliest DF (35.48), regardless of genotypes. Furthermore, both inorganic N and Mykovam greatly affected stem biomass (4.87 and 4.73 g plant^−1^, respectively), compared to nano-p and non-P application (4.5 and 4.2 g plant^−1^, respectively). This suggests that, similar to the trends observed on the effects of inorganic N (urea), inorganic P affected shoot traits notably and Mykovam had a comparable effect on stem biomass. Additionally, genotype Sbung seemed to have strong genotypic effect for the shoot traits examined.

#### 2.2.2. Chlorophyll Index

Similar to the other traits, different genotypes and P fertilizer sources showed significant effects (*p* < 0.05) and (*p* < 0.001) on leaf chlorophyll index at 28, 42, 56, and 70 days after growing (Table 8). The genotype 98C81 had the highest leaf chlorophyll index from 28 to 70 days after planting with 35.2, 42.8, 43.6, and 44.1 cm, respectively. This was followed by genotype Sbung with 33.5, 41.9, 42.7, and 42.9, respectively. The same genotypes were identified to have high chlorophyll index in Experiment 1, using different N fertilizer sources. All P fertilizer sources significantly increased chlorophyll index, compared to the control; however, the greatest effect was found in 50 kg P ha^−1^ inorganic P and nano-p. The interaction of genotypes and P fertilizer sources (*p* > 0.05) did not significantly influence the leaf chlorophyll index. Generally, the P source that showed strongest effects to shoot traits was inorganic P and Mykovam, and the effects of the other P source (nano-p) were higher than the control.

#### 2.2.3. Root Traits and Nodulation

Table 9 shows that genotypes and P sources significantly affected root traits, such as biomass, number of nodules, nodule dry weight, and shoot-to-root ratio (*p*< 0.01), while no interaction (GxP) was observed. All genotypes except ACS1 produced high root biomass, whereas only genotypes Sbung and 98C8 produced high number of nodules, compared to genotype ACS1 and Kaiabi, while genotype Hongry had the lowest. For nodule dry weight and shoot-to-root ratio, only Sbung (0.29 g plant^−1^) influenced the trait. On the other hand, all P fertilizer sources (50 kg ha^−1^ each inorganic P, Mykovam, and nano-p) increased root biomass, compared to the control, and were of similar effects (1.77, 1.76, and 1.55 g plant^−1^). Additionally, the highest number of nodules per plant was obtained at 50 kg P ha^−1^ of each inorganic P (14.81), nano-p (14.69), and Mykovam (14.65) application, while the lowest was obtained at 0 kg P ha^−1^ (12.03). Only inorganic P influenced the dry weight of nodules (0.27 g plant^−1^) and interaction of soybean genotypes x P fertilizer sources (GxP) insignificantly, in relation to root biomass, number of nodules per plant, and nodule dry weight. P and Mykovam had similar effects on nodule dry weight and root biomass, followed by nano-p, whereas non-P application had the lowest effects. All Mykovam, inorganic P, and nano-p increased root traits and nodulation, as compared to control in all genotypes.

#### 2.2.4. Yield Traits, Harvest Index, and Seed Protein

Table 10 presents the effects of genotypes and different P sources on the number of filled pods, total pods per plant, seeds per pod, 100–grain weight, seed yield, harvest index, and seed protein. All traits, except 100-grain weight, were significantly affected by genotypes, whereas all traits, except seeds per pod and 100-grain weight, were significantly influenced by P fertilizer sources. On the other hand, only significant interaction (GxP) was observed in seed protein. Genotype Sbung produced the highest number of filled pods (30.29), followed by 98C81 (28.91), while genotypes ACS1 and Hongry produced the lowest and similar number of filled pods (26.15 to 26.33). On the other hand, a higher total number of pods, seed weight, and harvest index were found in genotypes Sbung and 98C81. However, only genotype Sbung showed higher protein content (36.7%), compared to the rest of the genotypes. Moreover, the number of filled pods was significantly affected by Mykovam (29.53), followed by inorganic P (27.78) and nano-p (27.52), compared to the control (25.47), while the total number of pod and seed yields were significantly increased by all P sources (Mykovam, inorganic P, and nano-p). On the other hand, both inorganic P and nano-p had similar effects on harvest index and seed protein, followed by Mykovam, whereas non-P application had the lowest effects. Similar to N conditions, genotype Sbung had the high performance under P-conditions. Furthermore, all Mykovam, inorganic P, and nano-p increased yield, as compared to the control in all genotypes. However, there was GxP interaction on seed protein, where genotype Sbung had relatively higher seed protein, compared to other genotypes.

#### 2.2.5. Correlation Analyses among Yield and Yield Components

Seed yield in all treatments was positively and significantly correlated with the number of pods per plants (r = 0.67), 100-grain weight (r = 0.49), seed protein (r = 0.49), number of nodules (r = 0.38), shoot biomass (r = 0.33), and root biomass (r = 0.25) (Table 11). Additionally, a positive correlation of 100-grain weight and plant height (r = 0.45) was found. There was also significant correlation of days to flowering and plant height (r = 0.53), root biomass, and number of nodules (r = 0.43). Shoot biomass was positively correlated with the number of branches (r = 0.79), plant height (r = 0.60), and seed yield (r = 0.33). These results indicate that seed yield in all treatments was positively and significantly correlated with 100-grain weight and total number of pods per plant, which showed the difference in seed yield. Except plant height and number of branches, all the root, nodulation, and yield traits, as well as seed protein, were correlated significantly and positively to seed yield, indicating the development of effective and promising nodules of the crops, due to P supply, that could promote root growth, increase stem biomass, and nodule dry weight, which ultimately enhances the final yield.

#### 2.2.6. Path Analysis

Path analysis of direct and indirect effects of soybean character is shown in Table 12. Based on the results of path analysis, it is seen that number of nodules, root biomass, shoot biomass, number of pods per plant, and 100-grain weight give a positive direct effect on soybean yield. Therefore, selection of soybean genotypes for P sources must be upon changes of yield component. The most direct effect of character on seed yield was obtained from number of pods through seed yield (0.47), followed by 100-grain weight (0.41) and root biomass (0.28). Thus, the changes of character relationships in soybean, under P sources, is determined for variety selection.

## 3. Discussion

Seed yield and seed protein content of soybean genotypes varied in different N and P fertilizer sources. Regardless of N or P fertilizer sources, in the present study, genotype Sbung produced higher biomass, seed yields, and seed protein (Table 4 and Table 10). Significant differences among genotypes were also observed for days to flowering (Table 1 and Table 7). Genotypes ACS1, Hongry, and Kaiabi had longer duration to maturity and, thus, had lower chlorophyll index, compared to the other genotypes. However, these three genotypes had lower seed yield and harvest index. Machikowa and Laosuwan [21] reported that there was a tendency for later maturity genotypes to have lower harvest index, but the seed and biomass yield were variable and had strong genotype x environment interactions. Thus, the observed differences among genotypes could be attributed to the changes in seed protein content and their interaction with environment (GxN and GxP) [22]. Seed filling in soybean depends on carbon from current or stored assimilates transferred directly to the seed (particularly stem and leaves, which is influenced by genotypes, environment, and their interactions [23]. Moreover, significant differences were observed among genotypes for the total pods, 100-seed weight, and seed protein. When averaged across N and P sources with genotypes Sbung and 98C81 showing the highest performance (Table 4 and Table 10). Thus, the differences among soybean genotypes for seed yield and its components are highly related to genetic potential and their intrinsic characteristics [24]. Furthermore, among genotypes, only Sbung had the highest seed protein in both experiments, which may suggest that this is genetically controlled and seed protein is highly influenced by environment when N and P are available [25]. Enhanced plant growth and development was observed through chlorophyll index, root, and nodulation traits (number of nodules or dry weight) (Table 2, Table 3, Table 8 and Table 9) as the effect of different N and P sources. Those traits contributed to better high yield and seed protein content, which revealed the efficient N and P uptake [26]. Soybean is a very dynamic crop and has strong GxE in responses to the day length, which is the primary environmental variable triggering change in length of vegetable period, thus influencing greater seed setting than aboveground biomass yield. The increase of chlorophyll index can increase the photosynthetic rate of soybean, which attributed to the efficient use of light energy in transforming CO_2_ and inorganic N into plant materials, which ultimately increased seed yield and seed protein content.

Among N fertilizer sources, nano-n and urea produced higher biomass, seed yield, and seed protein. According to Manikandan and Subramanian [27], nano-n fertilizer absorbs on the surface of root, which allows nanoparticles cross the root surface cuticle and reach the root epidermis and enable precisely release nutrients in the root zone of plants by preventing rapid change in the chemical form of the nutrients in the soil, which can lead to high nutrient uptake, vigorous plant growth, and yield [28]. Additionally, N supplied from nodules and fertilizer is the most critical factor for maintaining the growth of photosynthetic organs and branching at the vegetative growth stage. *Rhizobium* helps to form root nodules and growth in vegetative organs, namely the stems, leaves, and roots with the availability of N elements; so, the leaves formed will also increase, which will increase the plant biomass [29]. Moreover, it increased in nodule weight, seed yield and seed protein. All genotypes responded positively to N as the soil at the experimental site were poor in N. Previous reports showed that the initial application of urea fertilizer at the rate of 80 kg N ha^−1^ significantly increased chlorophyll index and photosynthetic rate, which significantly impact growth, grain, and dry matter of soybean [30,31]. It was reported that urea fertilizer helps to promote plant growth, yield, and seed of soybean without a reduction in N fixation [32]. Moreover, the application of nano-n improved the growth and productivity of soybean, accumulation of dry matter and seed development which led to increase harvest index and seed protein of soybean through their large surface area and particle sizes less than the pore sizes of roots that can interact with some microorganisms that have symbiotic relationship with plant such as fungi and bacteria which can improve nutrient uptake by plants [28,33].

As previously reported, P is also an important factor that caused variability in plant height, number of pod/plants, 100-seed weight, seed yield and seed protein [34]. There were significant effects of genotype to soybean performance grown under P conditions regardless of sources. In this study, we found that application of inorganic P and nano-p at 50 kg P ha^−1^ increased seed yield, HI, and seed protein (Table 9). Application of P fertilizer can directly affect seed yield by influencing greater seed setting as well as aboveground biomass [35]. Soybean is a P-dependent crop, and application of P sources coordinated production, improved physiological characteristics. These results are consistent with previous findings those of Lui and Lal [36], who reported that availability of nano-p strongly influenced different parameters of plant productivity such as root, shoot length, number of reproductive buds, yield, and quality of soybean. The biomass of soybean was the foundation for the seed yield. Furthermore, our results are also in agreement with other reports by Salih et al. [37], who reported that proper application of different sources of P fertilizer improves nutrients supplied, which enhanced vegetative growth, affected pod and seed formation, increased plant biomass, and increased soybean yield. Thus, the pattern of distribution of biomass was a biological characteristic that depends on application of P like Mykovam. Phosphorous is important for the synthesis of chlorophyll in the chloroplasts, leaf photosynthesis, increased biomass production, root growth, nodulation, filled pods, seed yield, and N_2_ fixation [38]. Moreover, sources of P also enhance soil microbial function that support root growth and absorption of other macro and micronutrients. Our study showed that under different sources of P in the form of inorganic P and nano-p, dry matter production, seed yield and seed protein were enhanced.

## 4. Materials and Methods

Two separate, soybean-based experiments were set-up, according to the sources of nutrient, in order to analyze several parameters. Experiment 1 was conducted using different N fertilizer sources and Experiment 2 was performed using different P fertilizer sources. The screen house experiments were conducted in 2020 growing seasons at the Faculty of Agronomy, Royal University of Agriculture (RUA) in Cambodia (11°30′45.8″ N 104°54′03.2″ E). The location has a tropical climate, with a total monthly precipitation of 83.10 mm, that was high in June (in 2020) (Figure 1), and the total precipitation of the whole season of January to June was 175.10 mm and mean monthly temperature that ranged from 28.05 °C (in January) to 31.1 °C (in May) in 2020 (Figure 1).

### 4.1. Plant Materials

The five soybean genotypes used in the study consisted of three improved genotypes (98C81, ACS1, and Kaiabi) and two local genotypes (Sbung and Hongry), which these genotypes belong to X maturity group. The improved genotypes were obtained from the Conservation Agriculture Service Center (CASC), General Directorate of Agriculture (GDA) of Cambodia. These genotypes are very popular for researchers and farmers because of high yield (mean yield 2.1 t ha^−1^) and good grain of nutritional composition, all genotypes are 110-day maturity, as well as moderate tolerance to bacterial pustule (*Xanthomonas axonopodis* pv. *glycine*) and brown bean bug (*Riptotortus linearis*) [39]. The local genotypes, on the other hand, were collected from Ratanakiri province, and are widely used by local farmers in the country because of high yield (mean yield = 2.0 t ha^−1^) [39].

### 4.2. Experimental Design, Treatments and Management

To analyze more precisely the effects of the genotypes and N and/or P fertilizers, both experiments were arranged in two factorial randomized complete block design (RCBD) with four replications (three plants per pot and four pots per replication). The genotype was assigned as the factor A (Sbung, Hongry, 98C81, ACS1 and Kaiabi) in both experiments, while N sources (urea, nano-n, Nitroplus, and the control) were the factor B for experiment 1, whereas P sources (inorganic P, nano-p, Mykovam, and the control) in experiment 2.

Six pre-germinated seeds from each genotype were grown in a pot (height: 25 cm; diameter: 15 cm) filled with 8 kg of soil, taken from about 10 to 30 cm upper topsoil layer at the RUA experiment station. Prior to sowing, soil samples were collected at 0–40 cm depth, oven-dried, pulverized and analyzed for texture, potential of hydrogen (pH), chemical properties like organic matter (OM), total N, electrical conductivity (EC), available P and exchangeable K. The soils used in the two pot experiments were slightly acidic (6.8 pH), had low N (0.06%), very low EC (71.20 µS/cm), low available P (0.0005%), and very low exchangeable K (0.00009%) while the OM (2.4%) was medium with a soil texture of sandy loam. At 14 days after sowing (DAS), seedlings were thinned to three seedlings per pot and grown throughout the duration of the experiment.

For standard fertilizer rate application in this study, 90:50:27 kg ha^−1^ N-P_2_O_5_-K_2_O was applied following the recommended amount for the chemical fertilizer (urea, single super phosphate, and potassium chloride) application by the Cambodian Agricultural Research and Development Institute or CARDI [40]. Similar amount of nanofertilizers (FertiGroe^®^) was applied at a rate of 90:50:27 kg ha^−1^ N-P_2_O_5_-K_2_O. The P source used in experiment 1 was single super phosphate and the N source in experiment 2 was urea. The FertiGroe^®^ N and P nanofertilizers (herein referred to as nano-n or -p), developed by the Agricultural Systems Institute and Institute of Crop Science of the University of the Philippines Los Baños (UPLB), are inorganic single fertilizers in powder or granulated form [41]. Fertilizer application rates were calculated and converted into weight per pot, based on plant density as soil applied. All the nutrients, P, K, 50% N of urea, and nano-n were applied basally, while the remaining half of N was applied at 50% flowering as soil applied. The legume inoculant, Nitroplus, was used as a seed inoculant also at a rate equivalent to 90 kg N ha^−1^. On the other hand, the mycorrhizal fungi-based biofertilizer, Mykovam that served as the P source, was applied at a rate of 50 kg ha^−1^, and it is a soil-based biological fertilizer that was placed in a 5 cm deep hole at the center of pot. Nitroplus was applied by mixing with the seed before plating at the rate of 90 kg N ha^−1^. Both Nitroplus and Mykovam were developed in UPLB by the National Institute of Molecular Biology and Biotechnology (UPLB-BIOTECH).

For water management of soybean, the plants were irrigated as necessary using tap water. The amount of water applied was based on visual observation and if needed, the plants were irrigated up to two times per day.

### 4.3. Sampling and Measurements

#### 4.3.1. Growth Traits and Chlorophyll Index

Plant height was measured periodically (every 14 days) from the ground level to the first leaf of three plants randomly selected. At full pod stage [42], plants were cut at the ground, oven-dried at 60 °C for 72 h before weighing. Days to flowering (DF) and number of branches were determined when 50% of flowering occurred, based on three plants per pot. The chlorophyll index, on the other hand, was estimated using a portable chlorophyll meter (SPAD 502, Minolta Camera Co., Osaka, Japan). The two uppermost and fully developed leaves (on main braches) of three plants per pot were assessed for chlorophyll index, based on SPAD values.

#### 4.3.2. Root Traits and Nodulation

The number of nodules was determined at flowering stage. One of four pots (each pot has three plants) was randomly selected from each replicate, roots were extracted, and nodules were collected after washing gently using a fine sieve with running water to remove soil particles and organic debris. The average nodules per plant were counted after washing of nodules on each plant. The nodules were oven-dried at 45 °C for 12 h and weighed. Roots were cut after counting the nodule and oven-dried at 60 °C for 24 h and weighted. Shoot-to-root ratio was determined by dividing dry weigh of shoot biomass by dry weight of root biomass.

#### 4.3.3. Yield Traits, Harvest Index, and Seed Protein

Number of filled pods, total pods per plant and seeds per pod were determined from three pots and averaged for each treatment. The number of seeds per pod was counted by randomly selected 10 pods per pot. The seed yield, on the other hand, was measured for each pot and converted to yield in g plant^−1^, based on 13% grain moisture. One-hundred grain weight was counted and weighed separately using a sensitive scale. The relative proportion of seeds to the total shoot biomass was quantified as the harvest index (HI). The total N concentration of soybean seed was determined via the micro-Kjeldahl method, and the N concentration was multiplied by factor (6.25) to determine the total protein content [43].

### 4.4. Statistical Analysis

Analysis of variance (ANOVA) was performed to all the data gathered using the statistical analysis system (SAS version 9.1) [44]. Treatment means were compared using the least significant difference (LSD), at 5% and 1% level of significance. Correlation analysis was computed, and these coefficients were subjected to path analysis. There were five levels of factor A (genotypes) and four levels of factor B (fertilizer sources). For each factor, there were four replications and 180 observations (with in each replication), from which the means of each treatment were calculated.

## 5. Conclusions

Soybean genotypes varied in their responses under different N and P fertilizer sources. There were significant differences in growth, yield traits and seed protein among the genotypes. Notably, genotypes Sbung and 98C81 produced the highest total aboveground biomass and seed yield, whereas only genotype Sbung produced the highest seed protein content. Furthermore, the highest total aboveground biomass, leaf chlorophyll index, seed yield and seed protein were obtained under nano-n and urea, followed by Nitroplus conditions, applied with the same N rate (90 kg N ha^−1^), compared to non-N applied control. This was because the soils at the experimental site were poor in N. On the other hand, the highest seed yield was observed under different P-applied conditions with the same rate (50 kg ha^−1^) was produced under nano-p, Mykovam, and inorganic P, while seed protein was higher under nano-p and inorganic P conditions compared to Mykovam (biofertilizer) non-P applied conditions. These results indicate that different sources of N (nano-n, urea, and Nitroplus) and/or P (inorganic P, nano-p, and Mykovam) significantly increased shoot, root and nodulation traits, and seed yield as well as seed protein of soybean to different extents. Additionally, the nanofertilizers (nano-n or -p) and biofertilizers (Nitroplus and Mykovam) could be good alternatives to the standard N fertilizer (urea). Thus, aside from urea, the use of nanofertilizers (N or P) and *Rhizobium* inoculants (Nitroplus) in soybean production can improve soybean production in Cambodia. The results obtained in this research have potential application for increasing productivity of soybean, with the combinations of inorganic fertilizer, nanofertilzers, and Nitroplus. It is, therefore, necessary to further expand this research under different environmental conditions and farmer field conditions, under smallholder production systems, to evaluate yield increases, quantify environmental interaction, economics and potential for scaling and adoption.

## Figures and Tables

**Figure 1 plants-11-00298-f001:**
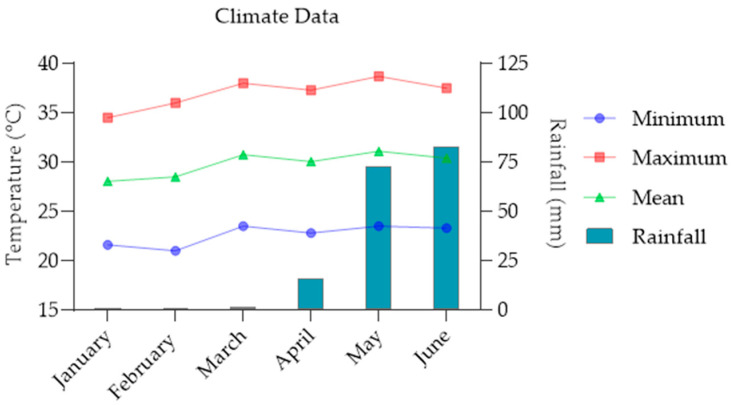
Maximum, minimum and mean temperatures and rainfall during the duration of the experiments (January–June 2020).

**Table 1 plants-11-00298-t001:** Plant height at different days after growing, days to flowering, number of branches, and stem biomass, as affected by genotypes and N fertilizer sources.

Factors	Plant Height (cm)			Days to Flowering	Number of Branches (no. plant^−1^)	Stem Biomass (g plant^−1^)
	28 Days	42 Days	56 Days			
Genotypes (G)						
Sbung	42.84 bc	82.67 b	89.16 b	36.09 b	2.61	5.39 a
Hongry	42.07 bc	83.19 b	87.49 b	35.18 b	2.46	4.79 c
98C81	45.93 ab	84.98 b	89.00 b	35.18 b	2.88	5.21 b
ACS1	49.37 a	91.94 a	94.43 a	41.03 a	2.89	4.50 d
Kaiabi	38.85 c	70.21 c	77.21 c	35.74 b	2.79	4.87 c
*p*-value	0.0016 **	0.0018 **	0.0025 **	0.0012 **	0.1450 ns	0.0022 **
N Fertilizers (N)						
Control	44.63 ab	83.81	87.94	36.57 b	2.81	4.54 c
Urea	46.08 a	81.14	87.49	36.20 b	2.62	5.50 a
Nano-N	42.34 b	81.89	88.00	37.34 a	2.81	4.92 b
Nitroplus	42.19 b	83.55	86.41	36.88 ab	2.67	4.86 b
*p*-value	0.0041 **	0.1125 ns	0.0981 ns	0.012 *	0.399 ns	0.0017 **
Interaction (GxN)						
*p*-value	0.0046 **	0.1330 ns	0.0015 **	0.0146 *	0.2257 ns	0.0455 *
CV (%)	7.28	4.62	2.95	3.97	12.03	6.39

Within column, means followed by the same lowercase letter/s do not differ significantly at LSD_0_._05_. *, ** significant at LSD_0_._05_ and LSD_0_._01_ level of probability, respectively; ns, do not differ significantly at 0.05 probability.

**Table 2 plants-11-00298-t002:** Leaf chlorophyll index at different days as affected by genotypes and N fertilizer sources.

Factors	28 Days	42 Days	56 Days	70 Days
Genotypes (G)				
Sbung	33.6 b	40.3 a	41.3 b	43.1 b
Hongry	31.5 c	38.7 c	39.5 c	41.7 c
98C81	35.6 a	40.4 a	41.7 a	43.5 a
ACS1	32.9 b	37.4 d	38.8 c	41.1 c
Kaiabi	33.2 b	39.3 b	41.1 b	42.6 b
*p*-value	0.0019 **	0.0023 **	0.0046 **	0.0012 **
N Fertilizers (N)				
Control	31.7 b	37.9 c	38.9 c	41.3 b
Urea	33.8 a	40.2 a	41.7 a	42.8 a
Nano-N	33.7 a	39.6 b	40.9 ab	43.5 a
Nitroplus	34.1 a	39.1 b	40.8 b	42.7 a
*p*-value	0.0016 **	0.0013 **	0.0006 ***	0.0025 **
Interaction (GxN)				
*p*-value	0.0821 ns	0.4112 ns	0.2413 ns	0.6542 ns
CV (%)	3.52	2.48	1.92	1.47

Within column, means followed by the same lowercase letter/s do not differ significantly at LSD_0_._05_.**, *** significant atLSD_0_._01_, and LSD_0_._001_ level of probability, respectively; ns, do not differ significantly at 0.05 probability. Leaf chlorophyll index was estimated by Soil Plant Analysis Development (SPAD) values.

**Table 3 plants-11-00298-t003:** Root biomass, number of nodules, nodule dry weight, and shoot to root ratio, as affected by genotypes and N fertilizer sources.

Factors	Root Biomass (g plant^−1^)	Number of Nodules (no. plant^−1^)	Nodule Dry Weight (g plant^−1^)	Shoot-to-Root (Ratio)
Genotypes (G)				
Sbung	7.51 a	34.25 a	0.42 a	2.51 a
Hongry	5.98 b	26.63 b	0.27 c	2.54 a
98C81	7.25 a	27.75 b	0.32 b	2.27 ab
ACS1	5.59 b	27.72 b	0.25 c	2.43 a
Kaiabi	7.04 a	23.50 c	0.24 c	2.13 b
*p*-value	0.001 ***	0.0056 **	0.0017 **	0.0046 **
N Fertilizers (N)				
Control	5.12 d	22.43 c	0.24 b	2.24 b
Urea	6.72 b	30.70 b	0.36 a	2.48 a
Nano-N	6.06 c	22.65 c	0.26 b	2.45 a
Nitroplus	8.78 a	32.58 a	0.35 a	2.34 ab
*p*-value	0.0061 **	0.0032 **	0.0014 **	0.0021 **
Interaction (GxN)				
*p*-value	0.3213 ns	0.0013 **	0.0016 **	0.0031 **
CV (%)	14.44	13.25	20.03	8.02

Within column, means followed by the same lowercase letter/s do not differ significantly at LSD_0_._05_. **, *** significant at LSD_0_._01_, and LSD_0_._001_ level of probability, respectively; ns, do not differ significantly at 0.05 probability.

**Table 4 plants-11-00298-t004:** Number of filled pods, total of pods, number of seed per pod, 100-grain weight, seed yield, harvest index, and seed protein, as affected by genotype and N fertilizer sources.

Factors	Filled Pods (no. plant^−1^)	Seed Per Pod (no. pod^−1^)	Total Pods (no plant^−1^)	100-Grain Weight (g)	Seed Yield (g plant^−1^)	Harvest Index (Ratio)	Seed Protein (%)
Genotype (G)							
Sbung	31.90 a	2.63 a	33.98 a	13.93 a	10.99 a	0.33 a	36.9 a
Hongry	27.61 b	2.48 b	31.16 b	12.87 b	9.60 b	0.29 b	35.5 c
98C81	32.99 a	2.66 a	34.65 a	13.69 a	10.63 a	0.32 a	35.9 b
ACS1	28.18 b	2.65 a	31.54 b	12.43 b	9.31 b	0.28 b	35.4 c
Kaiabi	29.02 b	2.66 a	31.91 b	13.08 b	9.77 b	0.29 b	35.7 bc
*p*-value	0.001 ***	0.0034 **	0.0022 **	0.0013 **	0.0014 **	0.001 ***	0.002 **
N Fertilizers (N)							
Control	27.46 c	2.51 b	30.56 c	12.73 b	9.57 b	0.29 c	35.0 c
Urea	31.90 a	2.65 a	34.19 a	13.47 a	10.14 ab	0.31 ab	36.4 a
Nano-N	30.60 b	2.65 a	32.81 b	13.40 a	10.61 a	0.32 a	36.4 a
Nitroplus	30.88 b	2.66 a	33.02 b	13.15 ab	9.92 b	0.30 bc	35.9 b
*p*-value	0.0023 **	0.0015 **	0.0017 **	0.0047 **	0.0076 **	0.0051 **	0.0016 **
Interaction (GxN)							
*p*-value	0.6411 ns	0.1725 ns	0.6141 ns	0.3096 ns	0.2265 ns	0.2114 ns	0.0436 *
CV (%)	3.82	5.29	3.67	7.43	7.21	5.54	2.03

Within column, means followed by the same lowercase letter/s do not differ significantly at LSD_0_._05_. *, **, *** significant at LSD_0_._05,_ LSD_0_._01_, and LSD_0_._001_ level of probability, respectively; ns, do not differ significantly at 0.05 probability.

**Table 5 plants-11-00298-t005:** Correlation coefficients analysis among seed yield (SY), plant height (PH), number of branches (NB), days to flowering (DF), number of nodules (NN), root biomass (RB), shoot biomass (SB), 100-grain weight (GW), total number of pods per plant (NPP), and seed protein (SP) of different soybean genotypes at different N fertilizer sources.

**Traits**	**SY**	**PH**	**NB**	**DF**
Seed yield, SY (g plant^−1^)	-			
Plant height, PH (cm)	0.12 ns	1.00		
Number of branches, NB	0.04 ns	−0.10 ns	1.00	
Days to flowering, DF	−0.12 ns	0.48 *	0.27 *	1.00
**Traits**	**SY**	**NN**	**RB**	**SB**
Seed yield, SY (g plant^−1^)	-			
Number of nodules, NN	0.26 *	1.00		
Root biomass, RB (g plant^−1^)	0.22 *	0.29 *	1.00	
Shoot biomass, SB (g plant^−1^)	0.04 ns	−0.01 ns	−0.07 ns	1.00
**Traits**	**SY**	**GW**	**NPP**	**SP**
Seed yield, SY (g plant^−1^)	-			
100-Grain weight, GW (g)	0.53 *	1.00		
Number of pods, NPP (plant^−1^)	0.39 *	0.51 *	1.00	
Seed protein, SP (%)	0.50 *	0.49 *	0.59 *	1.00

*, significant at 0.05 probability level of significance, respectively. ^ns^ do not differ significantly at 0.05 probability.

**Table 6 plants-11-00298-t006:** Path analysis of direct and indirect effects of soybean traits on seed yield (SY).

Traits	PH	NB	DF	NN	RB	SB	NPP	GW	SY
Plant height, PH (cm)	0.049	−0.009	−0.008	0.003	−0.018	0.012	−0.001	−0.02	0.008
Number of branches, NB	−0.004	0.095	−0.005	0.057	−0.015	−0.003	−0.005	−0.04	0.08
Days to flowering, DF	0.019	0.026	−0.019	0.076	−0.026	−0.007	−0.012	−0.11	−0.05
Number of nodules, NN	0.001	−0.02	0.005	−0.273	0.07	0.003	0.036	0.13	−0.048
Root biomass, RB (g plant^−1^)	−0.010	−0.013	0.004	−0.177	0.108	−0.004	0.035	0.104	0.047
Shoot biomass, SB (g plant^−1^)	0.039	−0.006	0.003	−0.014	−0.008	0.052	−0.002	0.003	0.067
Number of pods, NPP (Plant^−1^)	0.001	−0.008	0.03	−0.139	0.054	−0.003	0.070	0.177	0.182
100-Grain weight, GW (g)	−0.002	−0.001	0.006	−0.104	0.032	−0.001	0.036	0.34	0.306

**Table 7 plants-11-00298-t007:** Plant height at different days after growing, days to flowering, number of branches, and stem biomass, as affected by different genotypes and P fertilizer sources.

Factors	Plant Height (cm)			Days to Flowering	Number of Branches (no. plant^−1^)	Stem Biomass (g plant^−1^)
	28 Days	42 Days	56 Days			
Genotypes (G)						
Sbung	41.74 b	74.54 b	78.41 b	34.32 d	2.40	4.86 a
Hongry	39.68 b	75.82 b	79.26 b	35.19 c	2.22	4.54 b
98C81	41.30 b	75.39 b	77.70 b	34.79 cd	2.60	4.70 ab
ACS1	46.07 a	88.12 a	90.61 a	41.55 a	2.69	4.23 b
Kaiabi	34.05 c	65.72 c	68.91 c	36.77 b	2.66	4.60 b
*p*-value	0.0014 **	0.0016 **	0.0024 **	0.0017 **	0.0630 ns	0.0042 **
P Fertilizers (P)						
Mykovam	38.89 bc	75.81 a	78.54 b	36.43 b	2.62	4.73 ab
Inorganic-P	45.32 a	77.59 a	80.91 a	35.48 c	2.40	4.87 a
Nano-p	40.53 b	77.12 a	79.42 b	37.24 a	2.57	4.50 bc
Control	37.55 c	73.17 b	77.04 c	36.93 ab	2.46	4.20 c
*p*-value	0.0042 **	0.0141 *	0.0260 *	0.0350 *	0.2101 ns	0.0013 **
Interaction (GxP)						
*p*-value	0.0610 ns	0.8420 ns	0.0160 *	0.6412 ns	0.5240 ns	0.5913 ns
CV (%)	7.28	4.62	2.95	3.97	13.85	9.58

Within column, means followed by the same lowercase letter/s do not differ significantly at LSD_0_._05_. *, ** significant at LSD_0_._05_ and LSD_0_._01_ level of probability, respectively; ns, do not differ significantly at 0.05 probability.

**Table 8 plants-11-00298-t008:** Leaf chlorophyll index, as affected by genotypes and P fertilizer sources.

Factors	28 Days	42 Days	56 Days	70 Days
Genotypes (G)				
Sbung	33.5 b	41.9 b	42.7 b	42.9 b
Hongry	32.0 d	39.8 c	40.8 d	41.1 d
98C81	35.2 a	42.8 a	43.6 a	44.1 a
ACS1	33.1 bc	38.6 d	39.9 e	41.5 cd
Kaiabi	32.8 c	40.5 c	41.7 c	42.3 bc
*p*-value	0.0171 *	0.0013 **	0.0045 **	0.0012 **
P Fertilizers (P)				
Mykovam	33.7 ab	40.9 a	41.71 b	42.4 a
Inorganic-P	34.4 a	41.1 a	41.89 ab	42.7 a
Nano-P	33.5 b	41.4 a	42.27 a	42.6 a
Control	31.8 c	39.6 b	41.08 c	41.9 b
*p*-value	0.0027 **	0.0036 **	0.0241 *	0.0045 **
Interaction (GxP)				
*p*-value	0.3910 ns	0.5331 ns	0.2702 ns	0.3514 ns
CV (%)	3.52	2.48	1.92	1.47

Within column, means followed by the same lowercase letter do not differ significantly by LSD_0_._05_. *, ** significant at LSD_0_._05_ and LSD_0_._01_ level of probability, respectively; ns, do not differ significantly at 0.05 probability. Leaf chlorophyll index was estimated by soil plant analysis development (SPAD) values.

**Table 9 plants-11-00298-t009:** Root biomass, number of nodules, nodule dry weight, and shoot-to-root ratio, as affected by genotypes and P fertilizer sources.

Factors	Root Biomass (g plant^−1^)	Number of Nodules (no. plant^−1^)	Nodule Dry Weight (g plant^−1^)	Shoot-to-Root (Ratio)
Genotypes (G)				
Sbung	1.82 a	15.45 a	0.29 a	3.04 a
Hongry	1.59 a	12.72 c	0.22 b	2.71 b
98C81	1.62 a	14.94 a	0.22 b	2.80 ab
ACS1	1.31 b	13.47 b	0.20 b	2.63 b
Kaiabi	1.65 a	13.71 b	0.21 b	2.71 b
*p*-value	0.0023 **	0.0022 **	0.0312 *	0.0116 *
P Fertilizers (P)				
Mykovam	1.76 a	14.65 a	0.23 b	2.98 a
Inorganic-P	1.77 a	14.81 a	0.27 a	2.70 b
Nano-P	1.55 a	14.69 a	0.21 b	2.86 ab
Control	1.36 b	12.03 b	0.20 b	2.58 c
*p*-value	0.0051 **	0.0220 *	0.0132 *	0.0225 *
Interaction (GxP)				
*p*-value	0.3310 ns	0.1148 ns	0.0812 ns	0.0512 ns
CV (%)	14.44	13.25	16.02	9.57

Within column, means followed by the same lowercase letter/s do not differ significantly at LSD_0_._05_. *, ** significant at LSD_0_._05_ and LSD_0_._01_ level of probability, respectively; ns, do not differ significantly at 0.05 probability.

**Table 10 plants-11-00298-t010:** Number of filled pods, total of pods per plant, seeds per pod, 100-grain weight, seed yield, harvest index, and seed protein, as affected by genotypes and P fertilizer sources.

Factors	Filled Pods (no. plant^−1^)	Seed per Pod (no. pod^−1^)	Total Pods (no. plant^−1^)	100-Grain Weight (g)	Seed Yield (g plant^−1^)	Harvest Index (Ratio)	Seed Protein (%)
Genotype (G)							
Sbung	30.29 a	2.62 ab	30.34 a	13.51	9.15 a	0.28 a	36.7 a
Hongry	26.33 c	2.56 b	28.68 b	12.8	8.42 b	0.25 b	35.5 b
98C81	28.92 b	2.63 a	29.97 a	13.26	8.88 a	0.27 a	35.8 b
ACS1	26.15 c	2.65 a	29.36 ab	12.56	8.24 b	0.24 b	35.4 b
Kaiabi	26.19 c	2.63 a	29.27 ab	12.34	8.38 b	0.25 b	35.5 b
*p*-value	0.0241 *	0.0316 *	0.0044 **	0.1280 ns	0.0010 ***	0.0010 ***	0.0051 **
P Fertilizers (P)							
Mykovam	29.53 a	2.65	30.13 a	12.97	8.70 a	0.26 b	35.4 b
Inorganic-P	27.78 b	2.63	29.62 a	13.21	8.72 a	0.27 a	36.1 a
Nano-P	27.52 b	2.59	29.70 a	12.72	8.67 a	0.27 a	36.4 a
Control	25.47 c	2.59	28.65 b	12.68	8.26 b	0.25 c	34.9 c
P-value	0.0118 *	0.3820 ns	0.0081 **	0.2820 ns	0.0412 *	0.0015 **	0.0018 **
Interaction (GxP)							
*p*-value	0.6450 ns	0.1782 ns	0.6826 ns	0.3041 ns	0.2260 ns	0.214 ns	0.0450 *
CV (%)	3.82	5.29	3.67	7.43	7.21	5.54	2.76

Within columns, means followed by the same lowercase letter/s do not differ significantly at LSD_0_._05_. *, **, *** significant at LSD_0_._05_, LSD_0_._01_, and LSD_0_._001_ level of probability, respectively; ns, do not differ significantly at 0.05 probability.

**Table 11 plants-11-00298-t011:** Correlation coefficients analysis among seed yield (SY), plant height (PH), number of branches (NB), days to flowering (DF), number of nodules (NN), root biomass (RB), shoot biomass (SB), 100-grain weight (GW), total number of pods per plant (NPP), and seed protein (SP) of different soybean genotypes at different P fertilizer sources.

**Traits**	**SY**	**PH**	**NB**	**DF**
Seed yield, SY (g plant^−1^)	1.00			
Plant height, PH (cm)	−0.07 ns	1.00		
Number of branches, NB	−0.03 ns	0.04 ns	1.00	
Days to flowering, DF	0.13 ns	0.53 *	0.37 *	1.00
**Traits**	**SY**	**NN**	**RB**	**SB**
Seed yield, SY (g plant^−1^)	1.00			
Number of nodules, NN	0.38 *	1.00		
Root biomass, RB (g plant^−1^)	0.25 *	0.43 *	1.00	
Shoot biomass, SB (g plant^−1^)	0.33 *	0.60 *	0.79 *	1.00
**Traits**	**SY**	**GW**	**NPP**	**SP**
Seed yield, SY (g plant^−1^)	1.00			
100-Grain weight, GW (g)	0.49 *	1.00		
Number of pods, NPP (plant^−1^)	0.67 *	0.45 *	1.00	
Seed protein, SP (%)	0.49 *	0.22 ns	0.38 *	1.00

*, significant at 0.05 level of significance, respectively; ns, do not differ significantly at LSD_0_._05_.

**Table 12 plants-11-00298-t012:** Path analysis of direct and indirect effects of soybean characters on seed yield (SY).

Traits	PH	NB	DF	NN	RB	SB	NPP	GW	SY
Plant height, PH (cm)	−0.26	−0.01	0.08	0.02	0.06	−0.01	0.003	0.01	−0.17
Number of branches, NB	−0.01	−0.19	0.05	0.01	0.04	−0.01	0.08	−0.03	−0.03
Days to flowering, DF	−0.14	−0.07	0.15	−0.03	0.14	−0.11	−0.04	−0.07	−0.17
Number of nodules, NN	−0.05	−0.01	−0.05	0.09	−0.01	0.14	0.15	0.07	0.23
Root biomass, RB (g plant^−1^)	0.07	0.03	−0.09	0.04	−0.23	0.18	0.10	0.08	0.28
Shoot biomass, SB (g plant^−1^)	0.01	0.01	−0.07	0.05	−0.18	0.23	0.11	0.06	0.20
Number of pods, NPP (Plant^−1^)	0.002	−0.04	−0.02	0.04	−0.06	0.06	0.36	0.13	0.47
100-Grain weight, GW (g)	−0.01	0.02	−0.05	0.02	−0.07	0.05	0.16	0.28	0.41

## Data Availability

Not applicable.

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
