# Peer review of "Responses of Soybean Genotypes to Different Nitrogen and Phosphorus Sources: Impacts on Yield Components, Seed Yield, and Seed Protein"

_plants, 2022, doi:10.3390/plants11030298_

Round 1

Reviewer 1 Report

The work concerns effect of different genotypes and different nitrogen and phosphorus fertilizer sources on growth, seed yield and seed protein of soybean. The manuscript is clearly written and has the correct structure. The topic matches the journal’s scope.

The introduction provides a sufficiently broad and informative context. The authors convincingly define the originality of the research aims and motivate the investigations in the area of the topic. The objective is clearly defined.

The presented data is free from obvious errors. The results are concise and are separated from the discussion section. Results are well referenced with experimental data summarized in 12 tables. The discussion explores the significance of the results in the context of relevant published papers.

This is followed by a detailed description of experimental methods including Plant Materials, Experimental Design, Treatments and Management, Sampling and Measurements, Statistical Analysis. I recommend improving section 4.3.2 to add information about measured root traits. The authors focus on nodulation.

The screen house experiments were carried out in the 2020 growing seasons, which is a standard, validated approach in this type of study. It is somewhat regrettable, that it was not repeated in the following year, which would have made the study even more attractive. Statistical methods have been described with sufficient detail and were correctly applied.

The literature cited in the work is relevant to the study. The paper refers to 44 prior art publications, all of which are reasonably well referenced in the text. The conclusions of the study are presented in a short section highlighting the key results and their significance. 

Author Response

Please see the attached file the response to the reviewer's comments. 

Reviewer 2 Report

The manuscript is interesting and contains some valuable information. The presented study consists the original research results. It describes the effect of different N and P fertilizer sources on growth, seed yield and its components, as well as the seed protein content of five soybean genotypes. 

The title accurately reflect the content of the article.

The ‘Introduction’ part is clear. The research problem and the purpose of the work have been formulated correctly.

The ‘Results’ section is presented in a clear manner and generally correctly interpreted, but there are some minor errors, that must be addressed.

Line -106 – the title of section 2.1 is unclear.

Line 112-113 Significant differences among the genotypes (P<0.05) for plant height at 28, 42 and 56 112 days, DF and number of branches were observed.

For the number of branches, the differences between the genotypes were not statistically significant.

Line 116-117 Moreover, both genotypes ACS1 and 98C81 had the highest number of branches (2.89 and 2.88, respectively)…

The differences between the genotypes were not statistically significant.

Line 134 (P<0.05)

(P<0.01)

Line 137 (…) urea and Nitroplus yielded the highest leaf chlorophyll index (…)

higher

line 161-163 (…) nodule dry weight (0.35 g plant-1), and shoot-to-root ratio (2.48). The other N sources, urea and Nano-N, showed the effects 162 lower than Nitroplus but their effects were higher when compared to control.

Misinterpretation of data from Table 3. e.g. According to table 3 shoot-to-root ratio was 2.34 and it did not differ significantly from the control. Also nodule dry weight was similar on urea and Nitroplus.

line 173-173 Table 4 shows the effects of genotypes and different N sources on the number of filled pods, total pods per plant and number of seeds per pod 100-grain weight, seed yield per plant and harvest index.

And what about seed protein?

Line 177-178 (…) genotypes Sbung and 98C81 had the highest filled pods and total pods, whereas the genotypes Kaiabi, ACS1 and Hongry had the lowest.

(…) genotypes Sbung and 98C81 had higher filled pods and total pods, than the genotypes Kaiabi, ACS1 and Hongry.

Line 180-181 (…) genotypes Sbung and 98C81 also yielded the highest, whereas genotypes Kaiabi, Hongry and ACS1 had the lowest.

(…) genotypes Sbung and 98C81 also yielded higher, than genotypes Kaiabi, Hongry and ACS1.

Line 185-187 Application of all different N sources likewise had significant effects on the number of seeds per pod and 100-seed weight when compared to the control, with urea having the greatest effect, followed by Nano-N and Nitroplus.

Application of all different N sources likewise had significant effects on the number of seeds per pod and 100-seed weight when compared to the control.

Line 188-189 For seed yield, harvest index and seed protein, only urea and Nano-N had effects compared to Nitroplus and control.

According to table 4 there were no significant differences in seed weight and harvest index between urea effects and Nitroplus.

Line 213-214 shoot biomass (SH)

Shoot biomass (SB)

Table 6 – Days to flowering, DF, explain the abbreviation SY

Line 238 genotypes that had the earliest DF (34.32-36.77)

genotypes that had significantly earlier DF …

Line 255 (P<0.001)

(P<0.05)

Line 273-275 Table 9 shows that genotypes and P sources significantly affected root traits such as biomass, number of nodules and nodule dry weight (P< 0.01) while no interaction (GxP) was observed.

And, what about shoot-to-root ratio?

Line 287-288 Similar to N conditions, genotype Sbung had the high performance under P-conditions.

On what basis was this conclusion made, since there was no interaction?

Line 297-299 Table 10 presents the effects of genotypes and different P sources on the number of filled pods, total pods per plant, seeds per pod, 100–grain weight, seed yield and harvest index.

And, what about seed protein?

Line 303 (…) the rest of the genotypes produced the lowest and similar number of filled pods…

(…) the rest of the genotypes produced lower and similar number of filled pods…

Line 312-313 Similar to N conditions, genotype Sbung had the high performance under P-conditions.

On what basis was this conclusion made, since there was no interaction?

Line 322-335 Please, check this paragraph carefully, because the data does not agree with Table 11.

Table 11 seed protein (SP), 100-Seed weight

Materials and Methods part

Line 432 – (…) the total precipitation of the whole season of January to June was 285.10 mm

Is this correct? The figure 1 does not show such an amount of rainfall.

Line 462-464 The soils used in the two pot experiments were slightly acidic (6.8 pH), had low N (0.06%), very low EC (71.20 μS/cm), low available P (4.63 part per million) and very low exchangeable K (0.90 me/100 g soil).

Please, provide the content of all macronutrients (N, P, K) in the same units.

Line 515

Please provide the factors and the number of observations (n) from which the means for each treatment were calculated.

The 'Abstract' and ‘Conclusion’ sections should be check and eventually corrected and edited according to the suggested corrections in the ‘Results’ part.

The references are properly selected.

Author Response

Please see the attached file of the response to the reviewer's comments. Thank you.

Reviewer 3 Report

  1. “In Cambodia, like many developing countries, agriculture lands are predominately occupied by smallholder farmers, who have limited resources and capability (knowledge and technology) to manage the soils efficiently and optimize crop produc- tion.”. The application of Nano-N in the article increases soybean yield. However, Nano-N is more expensive than regular fertilizer, can Cambodia farmers afford it?
  2. The sentence “We hypothesized that there are differences in growth, yield traits and seed protein con- 101 tent of different soybean genotypes in response to different N and P fertilizer sources.”. It is suggested to delete this sentence, which is the result of speculation.
  3. In “Materials and Methods” section, the author did not write the relevant information on fertilizers, please add.
  4. Is the amount of fertilizer mentioned reasonable? Although soybeans have higher nitrogen requirements, they can fix nitrogen through root nodules. So, wonder if the author has considered this?
  5. In “4.2. Experimental Design, Treatments and Management” section, here we talk about specifying genotype A, which corresponds to five genotypes, so the number of treatments in the text is insufficient.
  6. “42 days 56 days Plant Height (cm)”. Supplemental significance analysis is recommended.
  7. The discussion did not analyze the reasons for the differences between treatments with different nitrogen sources and phosphorus sources, and it is suggested to supplement.

Author Response

Please see the attached file of the response to the reviewer's comment. Thank you.

Round 2

Reviewer 3 Report

Based on the author's revision, this article can be accepted.